# Halide Perovskites Films for Ionizing Radiation Detection: An Overview of Novel Solid-State Devices

**DOI:** 10.3390/s23104930

**Published:** 2023-05-20

**Authors:** Naomi Falsini, Alberto Ubaldini, Flavio Cicconi, Antonietta Rizzo, Anna Vinattieri, Mara Bruzzi

**Affiliations:** 1Nuclear Safety, Security and Sustainability Division, Fusion and Technology for Nuclear Safety and Security Department, Italian National Agency for New Technologies, Energy and Sustainable Economic Development (ENEA), Via Martiri di Monte Sole 4, 40129 Bologna, Italy; alberto.ubaldini@enea.it (A.U.); flavio.cicconi@enea.it (F.C.); antonietta.rizzo@enea.it (A.R.); 2Dipartimento di Fisica e Astronomia, Università degli Studi di Firenze, Via G. Sansone 1, 50019 Sesto Fiorentino, Italy; anna.vinattieri@unifi.it (A.V.); mara.bruzzi@unifi.it (M.B.); 3Istituto Nazionale di Fisica Nucleare–INFN, Sezione di Firenze, Via G. Sansone 1, 50019 Sesto Fiorentino, Italy

**Keywords:** halide perovskite, solid-state detectors, films, ionizing radiation, X-rays

## Abstract

Halide perovskites are a novel class of semiconductors that have attracted great interest in recent decades due to their peculiar properties of interest for optoelectronics. In fact, their use ranges from the field of sensors and light emitters to ionizing radiation detectors. Since 2015, ionizing radiation detectors exploiting perovskite films as active media have been developed. Recently, it has also been demonstrated that such devices can be suitable for medical and diagnostic applications. This review collects most of the recent and innovative publications regarding solid-state devices for the detection of X-rays, neutrons, and protons based on perovskite thin and thick films in order to show that this type of material can be used to design a new generation of devices and sensors. Thin and thick films of halide perovskites are indeed excellent candidates for low-cost and large-area device applications, where the film morphology allows the implementation on flexible devices, which is a cutting-edge topic in the sensor sector.

## 1. Introduction

Ionizing radiation refers to charged (α and β particles, and protons) and uncharged (neutrons) particles or electromagnetic radiation (X-rays and γ-rays) that have enough energy to ionize atoms or molecules by detaching electrons from them. Cosmic rays are also ionizing radiation. They are not a single class of particles, but a large group that contains, in addition to these, others not belonging to ordinary matter, such as muons or pions. However, cosmic rays will not be considered in this review. Neutrons are indirect ionizing radiation because they do not interact directly with electronic shells, but rather with atomic nuclei. However, several ionizing radiations are generated by these interactions so that matter ionization occurs via intermediate particles. In Figure 1 the most common ionizing particles are shown. Alpha particles or alpha rays are helium nuclei that have a big charge and mass. Therefore, they have a small capacity to penetrate matter and, for this reason, they mainly create many localized defects. Beta particles are electrons (β-) or positrons (β+) emitted by radioactive nucleides. They are about 7300 times lighter than the alpha particle, and therefore have a greater ability to penetrate matter. Along their path, they can generate γ and X-rays. High-energy photons (γ-rays and X-rays) have zero mass and therefore large penetration ability. Along their path in a material they can generate secondary particles and other lower energy photons. Protons belong to an intermediate case between alpha and beta particles, i.e., their mass is about four times smaller than that of alpha particles and therefore their penetration capacity is larger than that of alpha particles. They can generate secondary particles themselves. Neutrons differ from protons, although they have similar masses, because they have no electric charge and for this reason they have high penetration capacity, depending on their own energy. It is worth keeping in mind that four classes of neutrons are commonly classified based on their initial energy E${}_{i}$: thermal (E${}_{i}$ < 0.025 eV), epithermal (E${}_{i}$ 0.025 eV–100 keV), fast (E${}_{i}$ 100 keV–20 MeV), and high energy (E${}_{i}$ > 20 MeV).

In medical imaging and nuclear safety, X-rays, γ-rays, neutrons, and protons are used for non-destructive investigation of materials and biological tissues. In general, ionizing radiation is a widely used tool in a variety of applications, such as medical/diagnostic, environmental monitoring, and nuclear security [1,2,3].

In addition, many scientific areas, including nuclear materials, particle physics, materials science, structural chemistry, and molecular biology, make use of ionizing radiation for civilian and research purposes [4,5,6,7,8]. Moreover, the detection of these radiations is a topic of interest in many sectors, from industrial, medical and national security; as a consequence, research finalizing the development of radiation detectors is of high relevance. Radiation detection requires the ability to measure a physical effect in the detector itself. It can be a change in temperature, the light produced, or the electric charge induced. In this review, only the latter case will be dealt with. The most commonly used materials for such detection purposes are certainly semiconductors. In order to be used for a specific application, the detector must ensure certain requirements that will guarantee the achievement of specific objectives. The main characteristics of a detector are the energy range covered, the sensitivity, the noise, and, in most cases, the linear response. In addition, for specific applications, the detector has to be suitable for measurements under ambient conditions (room temperature…), as in the case of radiation exposure monitoring. In these cases, a flexible and portable device is an added value that also lends itself to a large range of applications. In fact, a flexible device is able to adapt itself to the geometry of the human body, useful for clinical radiotherapy. Moreover, for a reduction in the energy consumption, the device should be low-power, at high-efficiency, and low-cost.

In the field of ionizing radiation detectors, new semiconductor materials have appeared in recent years as candidates to replace traditional materials such as silicon and diamond. In particular, halide perovskites (HP), thanks to their promising properties, are conquering an ever-increasing space and are leading to significant progress in this sector. Halide perovskites possess many of the characteristics required for a material to be used as the active medium in a detector, i.e., they are defect-tolerant materials, have a high attenuation coefficient, and good carrier mobility. Furthermore, the preparation techniques are often simple, fast, and low cost, making perovskite detectors a strong competitor to traditional commercial detectors. Recently, much research has been focused on halide perovskite films. This is for multiple reasons. Perovskite single crystals have excellent photoelectric properties; however, they suffer limitations, such as the high manufacturing cost and the limited dimensions over a large area, making them not exploitable for industrial-scale production. On the other hand, a polycrystalline film allows a deposition over large areas, which is more economically advantageous. In addition, perovskite-based films can be the active medium of flexible devices [9,10,11,12].

Many review papers in the literature report the development of detectors based on perovskites single crystals for X-rays, γ-rays, and α particles [13,14,15]. Moreover, halide perovskites have opened the way to a novel generation of flexible and high spatial resolution detectors for in vivo and real-time applications. The applicability of perovskite films in the field of space exploration is also under investigation. In fact, perovskite solar cells are already actively being explored as a next-generation light-weight space photovoltaic technology [16] and recent works have demonstrated their radiation resistance in extreme environments typical of space [17].

The purpose of this review is to present the most recent results regarding detectors based on thin and thick films of halide perovskites in order to show their potentiality for designing a new generation of devices. In this manuscript, we will describe the preparation methods of perovskite films and show the main results obtained with organic and inorganic perovskite thick and thin films. The topics addressed in this review are depicted in Figure 2.

## 2. Halide Perovskites

In the last decade a significant effort of research in physics and chemistry has been directed to the development and study of halide perovskites that are one of the most promising classes of semiconductor materials for applications in the field of light generation (LEDs [18,19,20,21] and lasers [22]), renewables (solar cells), photonics [23], sensors, and more recently medical diagnostics [10,24,25,26]. HPs are materials described by the formula ABX_3_, where A can be an organic cation (CH_3_NH_3_^+^, NH_2_CH=NH_2_^+^) or an inorganic cation (Cs${}^{+}$, Rb${}^{+}$, …), B is a cation, such as Pb${}^{2+}$, Sn${}^{2+}$, Bi${}^{3+}$ …, and X is a halogen anion, that is Cl${}^{-}$, Br${}^{-}$, or I${}^{-}$. The crystalline structure of a generic perovskite is shown in Figure 2.

The interest in deeply investigating these semiconductors comes from their relevant electrical and optical properties, such as the direct bandgap that allows for efficient radiative recombination (in fact, in a direct bandgap semiconductor, the minimum of the conduction band and the maximum of the valence band occur at the same K-vector in the Brillouin zone), the gap tunability by changing the halogen and alloying [27,28,29], the high carrier mobility and the defect tolerance. In addition, low-cost methods can be used for the synthesis/deposition, unlike other semiconductors (Si, GaAs, Diamond, …etc.) commonly used for applications of similar interest. This is a crucial aspect because expensive processes are typically required for the fabrication of solid-state detectors. Another interesting aspect is that the material structure can be easily designed from bulk to ordered nano/microstructures to polycrystalline thick or thin films [30], depending on the specific protocol of growth. The final behavior of the device is related to the characteristics of the material structure.

The first halide perovskites to become famous were the hybrid ones, mainly based on methylammonium lead-trihalide (MAPbX${}_{3}$, with MA = CH${}_{3}$NH${}_{3}$, X = Cl, Br, I), because they were widely investigated to be used as the active medium of photovoltaic cells, with excellent results in the last years [19,31,32]. However, this kind of perovskite suffers from poor long-term stability (few thousands of hours [33,34]) due to the degradation of their organic component. The unstable nature of the organic component under light, moisture, high temperature, and other extreme conditions moved the research towards inorganic perovskites, which guarantee better stability over time while keeping the same optoelectronic properties.

The choice of the film (absorber) thickness is a compromise between mechanical flexibility and detector performance. In general, for a solid state detector the absorption capacity improves with thicker absorbers, but a thinner film allows for greater flexibility. This is true for high energy photons, but also for neutrons, because while thermal neutrons can give their energy entirely (and therefore be measured) even to a small thickness of sensitive material, fast ones can be entirely absorbed only by a great thickness. Although perovskite thin film devices can be easily fabricated, their performance could be inferior due to the limited radiation detection efficiency. Perovskite thick films are then more suitable for high-energy radiation detection. In fact, many groups have increased the thickness of the perovskite layer constituting the detector to optimize the collection of radiation so as to improve the performance of the device. Another aspect to consider in order to significantly improve the absorption of radiation is the control of the morphology and crystallinity of the film. However, in addition to the adequate (high) thickness of the film, it is necessary that the latter has an efficient charge transport in order to guarantee high performance of the device.

## 3. Techniques for Perovskite Films Preparation

Methods for perovskite film preparation are mainly classified into two categories: solution-based deposition techniques and solvent-free processes. Some techniques are more suitable for thick film deposition, others for thin films, and the preparation techniques are the same for both hybrid and inorganic perovskites. In addition, some techniques allow the layer thickness to be adjusted by controlling the deposition parameters.

Some of the techniques that we will describe below allow perovskite films to be deposited directly on the various substrates of the device or on the panel of the transistor array. This aspect is very advantageous because it avoids the difficulty of subsequently assembling the various films constituting the device.

### 3.1. Solution-Based Processes

Usually, in these techniques, perovskite powders (precursor salts) are mixed with a suitable solvent to obtain a solution with the required viscosity. After deposition, the solvent is allowed to evaporate or is somehow removed from the solution, depending on the type of technique. In order to facilitate the removal of the solvent, heat treatments are adopted in most cases. In general, the time that the solvent takes to evaporate/leave the solution is a relevant parameter because it determines the size of the perovskite crystallites that will get formed. Although these techniques are widely used even for large-scale production, a problem linked to the presence/removal of the solvent must be highlighted: the homogeneity of the crystalline film produced in terms of the size of the nano/microcrystals and the compactness of the film is not always guaranteed by this kind of technique. Usually, to overcome the reduced material quality, the addition of chemicals is required eventually in combination with post-deposition treatments as a thermal annealing in controlled atmosphere. Moreover, nano and micro-crystals, which are formed when solution-based techniques are used, present a high density of surface states that are efficient carrier traps limiting, in principle, the radiative recombination efficiency, negatively affecting the device performance, as the external quantum efficiencies for LED and shunt resistance in solar cells.

It should be noted that although both hybrid and inorganic HP films can be deposited by these techniques, solution-based processes cannot be used for the deposition of a specific kind of perovskite: CsPbCl${}_{3}$. To the best of our knowledge, the proper solvent for the deposition of a CsPbCl${}_{3}$ film by a solution-based method does not exist yet. The problem lies in the difficulty of finding a good solvent to dissolve both precursors salts (CsCl and PbCl${}_{2}$).

In the following, we itemize the most common solution-based techniques, also reported in Figure 3.

#### 3.1.1. Spin Coating

So far, spin coating represents the most common route for the deposition of perovskite-based films intended for optoelectronics and also for detectors [24,35]. The process is shown in Figure 3. This method consists of dropping the solution containing perovskite precursor salts onto the desired substrate, which is then accelerated through a rotating plate at a high angular velocity. Different types of substrate can be used, depending on the nature of the solvent (polar, non-polar…). After diffusion of the solution and evaporation of the solvent, a film is formed.

The uniformity of the film and its thickness depends on the deposition parameters, such as the rotation speed, but also on the concentration of the solution, its density and viscosity. A role is also played by the substrate wettability. In general, the quality of the thin films obtained by using such a technique is quite good, but the spin coating has the limitation of being poorly scalable on large substrates. Furthermore, the deposition of thick films is obtained by applying low rotation speeds, and this leads to poor reproducibility of this type of film. In 2015, Yakunin et al. [24] demonstrated the first perovskite thin-film X-ray detector, i.e., a p-i-n photodiode consisting of four spin-cast layers, where the active medium, a MAPbI_3_ film, ranged in thickness from 260 to 600 nm.

#### 3.1.2. Printing Techniques

Printing techniques are all the scalable evaporation deposition processes that can be used in mass production because they allow a large area to be covered. In the following, the most commonly used methods are listed.

Drop casting is a deposition technique in which a drop of solution is poured onto the desired substrate. The solvent is left to evaporate, the solute precipitates, and the formation of a microcrystalline film is obtained. It is a simple and effective method, and for this reason, it has been widely exploited to realize perovskite films to be integrated into ionizing radiation detectors. A recent example is the work realized by Bruzzi et al. [25] reporting a dosimeter for clinical radiotherapy consisting of a layer of microcrystalline inorganic perovskite of CsPbBr_3_ a few micrometers thick.Spray coating consists in the expulsion from a nozzle of small drops that form an aerosol by the use of an inert carrier gas [36]. The aerosol droplets hit the substrate and form homogeneous films when they dry. The thickness of the film thus created is controlled by varying the deposition parameters such as concentration, density, and pressure. The first research groups who were successful in obtaining X-ray detectors based on perovskite thick films were Yakunin et al. [24], who coated a glass substrate with MaPbI_3_-based film with a thickness in the range of 10–100 μm.Electro-spray deposition is one of the variants of spray coating. In this technique, small aerosol droplets are formed by Coulomb repulsion and are driven by the electrical gradient between the nozzle and the substrate. With this process, inorganic perovskite of Cs_2_TeI_6_ [37] was deposited in the form of a 25 μm-thick film used in an X-ray detector. The spraying parameters, i.e., the distance and the electric field between the nozzle and the substrate, and the substrate temperature have been optimized to properly control film thickness and morphology for its use as a high-energy radiation detector.Doctor blade involves the use of a blade as a deposition tool. It is the most widespread technique for large-scale production, because it has been adapted to roll-to-roll printing processes. The first large-area printable low-dose perovskite X-ray imaging device was demonstrated by Kim et al. [38], who in 2017 deposited through blade coating a polycrystalline 830 μm-thick MAPbI_3_ film.Inkjet printing is a technique where a drop of solution is ejected from a chamber reservoir through piezoelectric thermal actuators, and sent directly onto the substrate. Since the volume of material involved is small, this deposition technique is particularly suitable for the preparation of thin films. In addition, as the process occurs at a temperature below 100 ${}^{\circ }$C, it is also suitable for deposition on thin plastic substrates for flexible electronics. In the literature, X-ray detectors having as active medium both hybrid [11] and inorganic [12] perovskites printed by inkjet on a flexible substrate are present. Through the inkjet printing technique, Liu et al. [12] obtained the first 20 nm-thick quantum dots of CsPbBr_3_ perovskite deposited on a large-area PET substrate.

### 3.2. Solution-Free Processes

Solution-free processes are those methods where micro-crystalline powders of perovskite or its precursor salts are employed as the starting material for the layer fabrication. Perovskite films can, therefore, be prepared through pressing and melt methods or physical vapour deposition techniques, represented in Figure 4.

#### 3.2.1. Pressing and Melt Methods

These methods are used for manufacturing thick films and wafers, starting from the micro-crystalline powders of perovskite that are pressed and/or heated.

Pressing and hot pressing methods are typical powder molding techniques that allow
compact wafers of controlled shape and thickness to be obtained. They are manufacturing
processes that involve high isostatic pressure and/or high temperature to
reduce the porosity of the material and increase its density. Thick film/wafer-based ionizing radiation detectors have been effectively fabricated by means of these techniques. Shrestha et al. [39] reported in 2017 on high performance direct X-ray detectors based on sintered MAPbI_3_ perovskite wafers. By subjecting hybrid lead triiodide microcrystals to a pressure of 0.3 GPa for 5 min in a hydraulic press, they obtained a series of compact MAPbI_3_ wafers of thicknesses in the range from 0.2 to 1 mm, diameter of 1/2 inch with a low roughness (root mean square of 75 nm).More recently, however, Hu at al. fabricated a MAPbI_3_ wafer by the heat assisted high-pressure press method [40] employed for X-ray detection.Lead-free Cs_2_AgBiBr_6_ perovskite wafers with diameters up to 5 cm have been fabricated by Yang et al. using the isostatic-pressing method. They subsequently annealed the obtained material to promote the grains growth to the micrometric range [41].The hot pressing method was first proposed by Pan et al. [42] to fabricate inorganic perovskite CsPbBr_3_ films. The used method consists of placing the CsPbBr_3_ powder onto a fluorine-doped tin oxide (FTO) glass and heating at 873 K for 5 min. A melt perovskite layer is formed, which uniformly covers the glass substrate. Then, a pre-heated quartz is used to press the molten layer. After the slow decreasing of the temperature, the quartz plate is removed. By means of such a technique, a compact 240 μm thick quasi mono-crystalline film has been obtained with a single (101) orientation and few grain boundaries.Melt process was used by Matt et al. [43], who demonstrated direct X-ray detection by a thick film of crystalline CsPbBr_3_. Such a method involves the distribution of micro-crystalline CsPbBr_3_ onto FTO-covered grass substrate. The heating of the material up to 575 ${}^{\circ }C$ is reached. Then, the slow-cooling at room temperature gives rise to a 250 μm-thick film.

Although the promising performances of X-ray detectors based on perovskite thick films has been obtained with these techniques, the high temperatures involved in the processes are not compatible with plastic materials and therefore limit their use for fabricating devices on flexible substrates.

#### 3.2.2. Physical Vapour Deposition

This class of deposition methods is commonly used for fabricating thick films, but it is also particularly suitable for the deposition of thin films.

Close space sublimation is a type of deposition where the substrate and the source material are held close to each other in a vacuum chamber, which is pumped out. The source and the substrate are then heated. In detail, the source is heated to a fraction of its melting temperature and the substrate to a slightly lower temperature, causing the source to sublimate and allowing the vapors to travel the short distance from the substrate, where the condensation occurs producing a thin film. An entire cycle of such a process takes only a few minutes, making this technique very viable for large-scale production.Fernandez-Izquierdo et al. developed the confined space sublimation method for the deposition of stoichiometric thick films of CsPbBr_3_ [44].Radio-frequency magnetron sputtering is a vapour deposition where a target is bombarded with ions that cause its surface to be atomized. The target is constituted by perovskite salts or perovskite precursor salts finely grounded and pressed. On impact, the ions eject atoms/molecules from the surface, process known as sputtering. The ions causing the sputtering originate from a plasma generated near the target. Usually, the plasma is generated by applying a strong electric field near the target and, in the case of magnetron instruments, this plasma is confined and accelerated towards the target by a magnetic field. The sputtered material forms a vapour, which can be re-condensed on a substrate to form the film. The process allows the realization in controlled atmosphere of homogeneous polycrystalline films over a large area, with a nanometric control of thickness and the surface roughness [45,46,47], which makes the technique useful for large-scale applications, as required by the industry. The technique can be used for deposition on substrates of different materials, such as glass, metals, and even plastics for flexible devices. In addition, the growth can be performed at room temperature, limiting the strain between the substrate and the film. Another relevant advantage of the sputtering technique is the possibility of growing in vacuum directly in the chamber consecutive layers of different materials, that are necessary for the realization of prototype multilayer devices. Radio frequency (RF) magnetron sputtering was successfully employed by Bruzzi et al. for the first time to deposit a CsPbCl_3_ inorganic perovskite film as the active medium of a radiotherapy dosimeter [26]. In 2023, through the same technique, the first flexible CsPbCl_3_-based detector was demonstrated by the same group for real-time monitoring in proton therapy [9].A further variation of sputtering commonly referred to as reactive sputtering carries out this process with a background pressure of a reactive gas, such as oxygen or nitrogen to tune the composition of the resulting film.

## 4. Ionizing Radiation Detection

### 4.1. Radiation–Matter Interaction

Matter can interact with different types of particles, namely electromagnetic radiation (X-rays and γ-rays), charged particles (α and β particles, protons), and uncharged particles (neutrons). Thus the interaction of radiation with a material has to be described differently, depending on the type of particle involved in the interaction. In this review, devices for the detection of X-rays, neutrons, and protons are reported, therefore only the interactions of these particles with matter will be described. Moreover, only interactions with solid materials will be discussed here, because they are related to perovskite-based detectors.

An X-ray or γ-ray photon incident on the detector can interact with the electrons or with the nuclei of the material constituting the detector. The transfer of energy from the incident photon to the detector material can mainly occur in three different ways: Compton scattering, photoelectric absorption, and pair production [48], each of which is predominant in a certain energy range, which also depends on the atomic number of the element involved. Schematically, it can be said that the photoelectric effect is prevalent at relatively low energies, the Compton effect for intermediate energies and the pair production at higher energies. A schematic illustration of the interactions between X-rays and matter is shown in Figure 5.

The Compton effect occurs when the incident photon is scattered by the outer shell electron and a portion of the photon energy is transferred to the scattered electron. The complete absorption of the high-energy photon by an electron of the inner shell is the photoelectric absorption, since a photoelectron is produced with kinetic energy equal to the energy of the photon minus the binding energy of the electron. The production of a pair results in the generation of an electron-positron pair after the incident photon has been absorbed by the strong Coulomb electric field near the nuclei. Therefore, it can only occur when the energy of the photon is larger than the rest energy of the electron-positron pair, that is 1.022 MeV. Secondary electrons, including photoelectrons containing the full energy of the absorbed photons, scattered electrons, containing a fraction of the energy of the scattered photons, and electron-positron pairs can participate in a variety of both elastic and inelastic scattering events, which will convert their kinetic energy into a discrete number of electron–hole pairs. This number is calculated by dividing the total energy by the creation energies (in the range of a few eV) of the electron–hole pairs [49,50]. The cross sections of all three interactions increase with atomic number (Z). Therefore, high-Z materials are preferred, especially for the detection of high-energy X-rays and γ-rays which, in common detectors, can penetrate from millimeters to centimeters. The internal conversion is similar to the photoelectric effect: in this case the high-energy photon strips an electron from a deep shell from the atom. A second electron of a higher energy level fills the hole created by emitting a lower energy photon in the process.

There are many different interactions of neutrons with matter (reported in Figure 6a) and the probability of the interactions is strongly dependent on the neutron energy:Scattering: Neutrons may collide with nuclei and undergo either elastic or inelastic scattering. In the former case, a neutron transfers a fraction of its kinetic energy to the target nucleus without exciting the nucleus, whereas inelastic scattering occurs when the neutron transfers some of its kinetic energy to the target nucleus, by which the target becomes excited and the excitation energy is emitted as a γ-rays.Absorption: When a target nucleus absorbs a neutron, a wide range of nuclear reactions occur, including nuclear fission (this is more favorable at thermal energies): Radiative capture occurs when a neutron is absorbed by the target nucleus which then becomes excited and reaches stability by emission of electromagnetic energy in a form of γ-rays. Transmutation occurs when the target nucleus absorbs a neutron that results in an ejection of a charged particle such as a proton or an alpha particle. Nuclear fission reaction occurs when a fissile nucleus (this phenomenon concerns above all very heavy elements such as uranium or plutonium) splits into smaller nuclei (fission fragments). As the fission fragments are ejected, an average of 2.5 neutrons are emitted. In this way, a neutron can collide with another fissile nucleus which then splits into smaller nuclei and a reaction chain can take place, as in a nuclear power plant. Thermal neutrons are much more effective for the fission. In each scattering event, the neutrons lose some of their energy and eventually the fast neutrons slow down to being thermal neutrons.

Neutron detection mainly depends on elastic scattering with nuclei. Capture cross sections are small for most materials, with a single exception represented by thermal neutrons (E < 1 eV), which can be effectively captured by a few specific nuclei, such as ${}^{3}$He, ${}^{10}$B, ${}^{6}$Li and ${}^{157}$Gd, through nuclear reactions. These reactions produce secondary emissions which include protons, α and β particles and electromagnetic radiation [51]. Instead, in the case of fast neutrons, the interaction consists of elastic diffusion and since the hydrogen nucleus has the same mass as the neutron, it becomes the most effective energy degrader. The recoil nucleus, which in the case of hydrogen is the proton, transfers its energy to the surrounding electrons in a short range, similarly as an α particle.

Protons can also interact with matter in many ways with rather complex mechanisms (Figure 6b). It must be taken into account in this case that they are charged particles and therefore they can interact with both atomic nuclei and electron shells. Even proton Bremsstrahlung is theoretically possible, but generally is negligible. They can act directly as ionizing particle via inelastic Coulombic interactions, losing energy that releases an electron; they can undergo to a deflection of their trajectory by repulsive Coulomb elastic scattering with nucleus; they can have non-elastic nuclear interaction with the formation of secondary particles, such as neutrons, γ-rays and lower energy protons.

Proton therapy offers a substantial clinical advantage over conventional photon therapy, due to the unique depth-dose characteristics of protons, which can be exploited to achieve significant reductions in normal tissue doses proximal and distal to the target volume. These may allow a more accurate sparing of normal tissues, potentially improving local control and survival. In contrast to photons, when protons of a given energy (typically in the range of 70 to 250 MeV) penetrate matter, they slow down continuously as a function of depth. The rate of their energy loss (called “linear energy transfer” or LET) increases with decreasing velocity. This continues until their entire energy is depleted and then they come to an abrupt stop. This process of dose deposition produces a characteristic depth-dose curve called “Bragg curve”: the point of highest dose corresponds to the Bragg peak, the depth of the peak, the proton range, is a function of the initial energy. By tuning the energy of the beam, the Bragg peak may be focused at the tumor, to improve spatial accuracy in dose delivery and thus spare healthy surrounding tissues. Dosimetric systems in proton therapy are used to support treatment plans, by taking into account parameters as proton fluxes, dose-rates, beam angles, and patient positioning aspects required to deliver an appropriate dose to the target volume.

### 4.2. Radiation Detection

Radiation detection relies on partial or complete energy transfer from photons/particles to the detector material. The operating principle of all radiation detectors is the same: they convert radiation into optical or electrical signals which can subsequently be amplified and processed by conventional electronics. In addition to specific radiation requirements, detectors must meet some common requirements: they must have a high absorption cross section, as well as a high charge carrier/photon conversion efficiency, low noise, high spatial and timing resolutions, and high radiation hardness.

Based on their operating mechanisms, radiation detectors can be classified into different categories: solid-state detectors, i.e., with direct conversion, scintillators, liquefied noble gas detectors, and gaseous ionization detectors, which are indirect conversion detectors. In this review only the first category will be discussed.

### 4.3. Radiation Detectors: Brief Overview

Solid-state detectors provide a signal by collecting the current generated by the charge liberated in the passage of the particle through a semiconductor. As shown in Figure 7a, semiconductors can be intrinsic, p-doped, or n-doped (if some atoms in the lattice are replaced by hetero-elements with fewer or more electrons than the parent element, respectively). For the function of a detector, high purity materials are required. At low temperature, from an electronic point of view, the system is described by the scheme of the semiconductor band structure of Figure 7b. A high-energy photon can excite an electron, making it jump from the valence band to the conduction one (Figure 7c). In this way an electron–hole pair is generated, capable of carrying electric current. Electrons and holes have opposite polarities so they move in opposite directions under the action of an electric field. Under the action of gamma rays, numerous electron–hole pairs are generated in the semiconductor. A picture showing the operating mechanisms of a X-ray detector is reported in Figure 7d. The peak current that can be measured is proportional to the intensity of the gamma ray beam.

The operating mechanisms of a solid-state detector for neutrons and protons is shown in Figure 8. Neutrons have a higher penetration capacity than protons, but they do not directly ionize matter. Only particles that are generated due to one of the nuclear processes described above induce ionization. A cloud of electron and hole particles is produced, initially it is more localized (Figure 8a). Instead, the effects of a beam of protons on matter are numerous, as previously described. They can directly ionize matter along their path or generate a nuclear process. As a consequence, a cloud of electrons and holes can be created, however it is very localized, due to the limited mean free path of the protons (Figure 8b).

In a solid-state detector the electron–hole pairs generated by the incident photons or particles induce an electrical signal at electrodes. As will be explained in more detail in the next paragraph, to facilitate charge transport and maximize charge collection efficiency, detector materials should have the so-called “mobility-lifetime product” (μτ product) large enough to allow electrons or holes to travel through the detector and get to the contacts. Furthermore, to minimize the leakage current under a large electric field, the bandgap of the detector materials must be relatively large and the defect densities should be opportunely controlled along the entire device structure.

A basic radiation detection chain is shown in Figure 9.

Incident radiation releases energy in the sensing material. The conversion of the energy deposited by radiation to an electrical signal can be achieved in different ways. In the case of a semiconductor sensor, electron–hole pairs are created in a number proportional to the absorbed energy. To establish the electric field, a potential is applied between the electrodes, to accelerate the generated charge carriers. They are swept by the applied electric field causing an induced charge on the opposite electrodes, as predicted by the Shockley–Ramo theorem [52,53]. This low-level electrical signal is initially sent to a pre-amplifier stage, which has also the function of integrating the signal; it is then fed to a pulse-shaper and finally it is digitized for storage and analysis.

Although the primary physical quantity measuring the signal energy is the charge, when the pulse shape is the same for all signal magnitudes, the pulse amplitude, called “pulse height” may be considered a more convenient parameter. This “pulse height analysis”, gives as output the pulse height spectrum, or energy spectrum.

The detector signal may be collected either in pulse- or current-mode, depending on the specific application. The output of the pulse-mode detector is a train of voltage pulses, each corresponding to a detected particle: it may be thus regarded as a single-event process. The voltage signal is then amplified and collected as a histogram through a shaping amplifier and a multichannel analyzer. Under current-mode operation, instead, during irradiation the current signal at a constant applied voltage is integrated on selected sampling intervals and read-out within fixed intervals as a function of time.

The performance of a radiation detector in general is measured in terms of parameters as gain, response time, signal to noise ratio, sensitivity.

In general, the read-out charge signal $Q_{s}$ is proportional to the absorbed energy *E* through the relationship (Equation 1):(1)$$Q_{s}=qE/E_{i}$$
where *E* is the absorbed energy, $E_{i}$ the energy required to form a charge pair, and *q* is the electronic charge. The ionization energy $E_{i}$ is proportional to the bandgap, $E_{g}$, so that higher bandgap materials yield less signal charge. The relationship between $E_{i}$ and $E_{g}$ has been studied extensively in literature. Klein [54] calculated the average amount of radiation energy consumed per pair as a sum of three contributions: the bandgap, optical phonon losses, and the residual kinetic energy as $\frac{9}{5}E_{g}$. Nonetheless, $E_{i}$ mean-ionization energies smaller than those predicted by the Klein’s relationship have been observed in various halide semiconductors as e.g., perovskites [49]. An empirical model developed by Devanathan et al. [55] includes these deviations suggesting the following relationship (Equation 2) of $E_{i}$ vs bandgap:(2)$$E_{i}=2E_{g}+1.43e\;V$$

The sensitivity per unit active volume of the semiconductor materials is then defined as: $S_{v}=q\rho /E_{i}$, with ρ mass density [56]. In general, lower bandgap semiconductor materials are characterized by lower ionization energies, thus providing higher concentrations of generated e–h pairs and consequently higher signals. On the other hand, higher bandgap semiconductors allow for a negligible thermal generation of e–h pairs, thus a reduced noise from leakage current during exposure to radiation.

For this reason, semiconductors with intermediate gaps, typically in the range 1–3 eV are considered, as e.g., Si and SiC, with sensitivity per unit volume of 637 nC${\mathrm{Gy}}^{-1}$${\mathrm{mm}}^{-3}$ and 411 nC${\mathrm{Gy}}^{-1}$${\mathrm{mm}}^{-3}$, respectively, where Gy is the unit of dose ( 1 Gy= 1 J/kg). In a detector working as a dosimeter, an important requirement is that the collected charge/current signal should be linear functions of the dose/dose-rate. The slope of such linear functions is the sensitivity of the device *S*, expressed usually in nC/Gy. In segmented dosimeters, used to imaging the dose on a bidimensional map, it is also quite important to evidence the sensitivity per unit active area for each pixel, a parameter expressed as nC${\mathrm{Gy}}^{-1}$${\mathrm{mm}}^{-2}$.

The semiconductor sensor may operate either as a photoresistance or as a p-n device. The latter structure is particularly useful to achieve high signal-to-noise ratios and fast responses, due to the presence of a depleted region at the interface of the two differently doped regions. This is especially important when the gap of the material is low, as in Ge or Si, as the thermally generated free carrier concentrations in the semiconductor are of the same order or higher than the concentration of e–h pairs generated by the impinging radiation. Assuming a photodiode under partial depletion, and a radiation-induced uniform generation of e–h pairs throughout the semiconductor, those carriers generated in the junction region will be swept across the junction by the electric field and collected within a few nanoseconds. Charged carriers created outside the depletion region will cause a transient increase in the local carrier density, giving rise to an adding current contribution. This will appear as a delayed component to the current signal, due to the longer time required for carriers to diffuse to the junction from the neutral bulk. This contribution is in fact proportional to the minority diffusion length $L=\sqrt{D\tau }$, with *D* diffusivity and τ minority carrier lifetime. To fasten and maximize the signal it is thus convenient to increase the depleted depth of the photodiode up to the physical total thickness of the device by increasing the externally applied reverse voltage (this practice, in case of high doping and thin devices, may actually be limited by unwanted breakdown effects.).

Conversely, a photoconductor consists of a slab of semiconductor, in bulk or thin-film form, with ohmic contacts either at opposite ends or interdigitated. When incident radiation falls on the surface of the photoconductor, carriers are generated, resulting in an increase of the electrical conductivity. Considering a steady flow of photons impinging uniformly on the surface of an extrinsic photoconductor with intercontact distance *d*, the photocurrent gain of the device is defined [57] as:(3)$$\gamma =\frac{\mu \tau \varepsilon }{d}$$
where μ is the carrier mobility, τ is the carrier lifetime and ε the applied electric field. In a semiconductor, the total induced charge is given by the sum of the induced charges due both to the electrons and holes. For a detector having a uniform electric field *E* there are two relevant μτ products for holes and electrons. We observe that in compound semiconductors, the $\mu_{h}\tau_{h}$ term is usually much lower than the corresponding $\mu_{n}\tau_{n}$ for electrons [52]. The responsivity of the extrinsic photoconductor, namely the ratio of the generated current and the incident radiation power, is then directly proportional to this photoconductive gain, which is actually given by the ratio of the drift length to the device intercontact length, *d*.

This ability of photoconductors to exhibit a photoconductive gain, so that the number of charge carriers that flow through the external circuit in response to the input signal can exceed the number of electron hole pairs generated by photon absorption, has drawbacks in longer temporal responses and potentially higher noise contributions with respect to photodiodes. In fact, the temporal response of this kind of device is directly related to the lifetime τ of the photogenerated e–h pairs. Moreover, the absence of a depleted region due to the ohmic configuration, may give rise to a non-negligible dark conductivity, due to the thermal generation of charged carriers, which may be particularly detrimental in small-bandgap materials.

For direct X-rays detectors, sensitivity is generally governed by X-ray attenuation, electron–hole pair generation, carrier extraction and photoconductive gain. For a given material, the capacity of attenuation and electron–hole pair generation energy are fixed, and thus the sensitivity is mainly dependent on carrier extraction and gain. The ability of carrier extraction can be determined by the mobility-lifetime (μτ) product, which can be derived by fitting the photoconductivity using the Hecht Equation (Equation 4) [58]:(4)$$I=\frac{I_{0}\mu \tau V}{L^{2}}\left(1-exp\left(-\frac{L^{2}}{\mu \tau V}\right)\right)$$
where $I_{0}$ is the saturated photocurrent, *L* is the thickness, *V* is the applied bias, *t* is the carrier lifetime. We observe that this formulation is valid in case of transport properties and electric field uniform in the whole volume of the sensor. This situation may be in practice considered when using either a resistive device or a reverse biased p-i-n detector [59]. Modified Hecht’s expression for semi-insulating semiconductors where space charges may be present have been also reported, which apply in case native and radiation induced defects are influencing the carriers transport properties [60,61,62]. In this case, a generalized Hecht equation is considered, incorporating both types of carriers, starting from the Gunn theorem [63].

Photoconductivity gain is desirable for highly sensitive photodetectors. In principle, photodetectors with diode structures such as p–n junction, p–i–n junction, or Schottky junction diodes have no intrinsic gain. Under large bias voltage, there may be impact ionization induced avalanche gain, which can be ascribed to the accelerated electrons or holes colliding with atomic electron, usually happening under large electric field. This impact ionization effect has been extensively applied to increase the performances in conventional semiconductor detectors [57]. However, semiconductors characterized by a heavily disordered lattice do not normally exhibit impact ionization. In fact, here carriers are more localized and charge transport occurs mainly through thermally activated hopping. As a result, it is difficult for carriers to accumulate sufficient kinetic energy to enable an avalanche effect. Nonetheless, a photoconductive gain may be observed indeed, which may originate by trapped carriers at defects, improving tunnelling injection at electrodes [64]. When minority charges are trapped at defects or interfaces, they induce the activation of a net fixed charge. Thus, more majority carriers are injected from electrodes in order to maintain the charge neutrality into the material. A net multiplication effect may occur, with several charges flowing for every charged pair created by the initial excitation. The total multiplication factor is controlled by the fraction between traps’ relaxation time and carriers’ transit time [65]. Thus, detection sensitivity may be in principle improved by increasing gain through introducing defects. Unfortunately, this procedure has drawbacks in the increased time responses and dark currents. In fact, gain is usually observed when the carrier lifetime exceeds the carrier transit time, and thus high gain is often associated with long carrier trapping, which results in a slower detector response. Moreover, defects are associated to the appearance of deep energy levels within the forbidden gap, which may act as thermal generators of carriers, in turn increasing the dark current of the device. Similarly, in flexible detectors enhanced X-rays responses have been ascribed to the activation of a photoconductive gain process assisted by defective states induced by the mechanical strains, interpretation actually supported by slower response times [66].

### 4.4. Perovskite Detectors For X-rays

The interest in perovskite-based ionizing radiation detectors is quite recent, also considering that the first perovskite film-based X-rays detector was published in 2015 by Yakunin et al. [24]. Since then, research has aimed to increase the thickness of the film, i.e., the absorption of the radiation, while maintaining a high μτ in order to guarantee a high charge collection efficiency. Pressing and melting processes were often employed to obtain thick films and wafers [39,40,42,43]. Hu et al. [40] measured a high value of sensitivity per unit area of $1.22\times 10^{5}$ μC${\mathrm{Gy}}^{-1}$${\mathrm{cm}}^{-2}$ in a perovskite film detector consisting of an 800 μm-thick MAPbI${}_{3}$ wafer obtained by heat-assisted pressing. Kim et al. [38], instead, using a solution-based technique, fabricated a highly efficient large-area MAPbI${}_{3}$-based X-ray detector with thickness of 830 μm, allowing printing on a thin film transistor backplane of large area. The authors report a sensitivity per unit area of up to $1.1\times 10^{4}$ μC${\mathrm{Gy}}^{-1}$${\mathrm{cm}}^{-2}$, a large value of the product μτ of the charge carrier ($1.0\times 10^{-4}$ cm${}^{2}$$V^{-1}$) and evaluated a rapid response to a train of short X-ray pulses with a pulse width of 50 ms. The highest value of sensitivity per unit area ($4.2\times 10^{5}$ μC${\mathrm{Gy}}^{-1}$${\mathrm{cm}}^{-2}$) was achieved by Xiao et al. [67], after grain engineering. The active medium of such ultra-sensitive perovskite X-ray detector was a 1.3 mm thick MAPbI${}_{3}$ wafer obtained through hot pressing method.

In the last few years the research focused on the investigation of new perovskite stoichiometries [37,68], deposition techniques as well as device structures [11], that can ensure high detection performances. Relevant parameters of all recent solid-state devices for X-ray detection based on thick films and wafers are shown in Table 1 for organic perovskites and in Table 2 for inorganic and multiple-cation perovskites. Flexible devices and lead free perovskites are also indicated.

Apart from a few cases of flexible devices consisting of a perovskite thick film [11,65,73], in which the film thicknesses were anyhow less than a few tens of micrometres, or perovskite-filled membranes [74], thick films or wafers generally did not lead themselves to the implementation of flexible devices. For this reason, many research works have recently focused on the synthesis and use of thin films as active media of radiation detectors. The first perovskite-based flexible X-ray detectors were demonstrated by Liu et al. [12] in 2019. They printed a inorganic CsPbBr${}_{3}$ quantum dots (QDs) solution onto pre-patterned metal electrodes on Si or flexible substrate in PET through the inkjet printing method. They obtained a photoconductor made of a 20 nm-thick perovskite layer. The authors observed a rapid response (rise time of about 30 ms) to X-rays at bias voltages below 1 V. The sensitivity value achieved by the flexible device was 17.7 μC${\mathrm{Gy}}^{-1}$${\mathrm{cm}}^{-2}$ under soft X-ray-near UV synchrotron beamline (100–2500) eV, biased at 0.1 V, while the one for the rigid device was 83 μC${\mathrm{Gy}}^{-1}$${\mathrm{cm}}^{-2}$ under same irradiation conditions. Despite the good X-ray detection performance, the photoconductor architecture and the QDs suspension brought trouble to material stabilization and transport control.

In Table 3 all recent solid-state devices for X-ray detection based on thin films of perovskites are reported, to the best of our knowledge, with indication of flexible devices.

Device structures and photos for flexible X-rays detectors are shown in Figure 10.

Halide perovskite film-based detectors for ionizing radiation have seen rapid development in recent years, especially for neutron and proton detection in the last couple of years. The advantages of halide perovskites are tunable compositions to meet the requirements of different types of radiation, low-cost and versatile deposition techniques for large-scale production, and excellent optoelectronic properties for efficient detection. Since 2015, the year of the development of the first detector for X-rays based on organic perovskite (MAPbI${}_{3}$), in order to increase the device efficiency. The first research focused on the investigation of techniques to increase the thickness. Then, new stoichiometries, also lead-free, and novel device architectures have been investigated. Recently, research has moved towards the study of flexible substrate detectors. Despite the promising results shown above, many efforts still need to be made to obtain high values of sensitivity per unit area of a detector implemented on a flexible substrate.

### 4.5. Perovskite Detectors for Neutrons

Solid-state neutron detection devices found in the literature are almost exclusively single crystals. However, Fernandez-Izquierdo et al. developed a cesium lead bromide (CsPbBr${}_{3}$) thin-film-based indirect-conversion solid-state thermal neutron detector, where the perovskite thin film is combined with a ${}^{10}$B conversion film [44] (Figure 11a). In that work, the close space sublimation (CSS) method was used for the deposition of the perovskite film. The CSS process used small crystals as precursors to enable films with large columnar grain growth across the entire film thickness of 8–9 μm. The mechanism of detection is indirect, the neutron response is enabled by the low leakage current (around 10${}^{-8}$ A mm${}^{-2}$) and fast response of the Ga${}_{2}$O${}_{3}$/CsPbBr${}_{3}$ diode. The efficiency reported for thermal neutrons is around 1%. By doping the surface of the perovskite layer with Cl anions coming from the treatment with PbCl${}_{2}$ vapor, such efficiency value was further improved up to 2.5%. The same group reported an optimized microstructured material (the ${}^{10}$B conversion layer was backfilled into the microstructures etched on the CsPbBr${}_{3}$ film, shown in Figure 11b) for thermal neutron detection with a combination of different techniques, such as a solvent-free thin-film deposition, perovskite patterning and dry etching process using HBr + Ar plasma. The fabricated microstructured CsPbBr${}_{3}$ thermal neutron detectors showed an efficiency of 4.3% [83]. In Figure 11c the neutron detection efficiency is shown as a function of the microstructure depth. As explained by Caraveo-Frescas et al., the different values of experimental neutron detection efficiency compared with theoretical efficiency can be related to two causes: the packing density of ${}^{10}$B in the microstructures, since a 100% filling density is unlikely to be experimentally achieved, and the surface defects generated during dry etching that can negatively affect the charge collection efficiency of the neutron detector [84]. The main performances of devices based on perovskite films for the detection of thermal neutrons are reported in Table 4.

### 4.6. Perovskite Detectors for Protons

Very recently, frontier research on perovskite detectors for ionizing radiation also moved towards proton therapy flux monitoring and dosimetry applications. Thin film prototype detectors for proton flux monitoring based on CsPbCl${}_{3}$ 1 μm-thick films grown with room temperature RF magnetron sputtering on flexible substrates, 125 μm thick, equipped with Pd interdigitated electrodes (IDEs) with 100 μm width/contacts pitch have been reported in Ref. [9]. A photo of the device developed for this purpose is shown in Figure 12a. Bruzzi et al. report the first measurements in real-time configuration with a reverse bias of 2V under proton beams with energy in the range 100–228 MeV and 1–10 nA extraction currents, of interest for proton therapy applications. Experimental results evidence the good performances of the CsPbCl${}_{3}$/Pd IDE in terms of real-time monitoring, linearity with current extraction and proton fluxes in almost two order of magnitude range, $4\times 10^{7}$–$2\times 10^{9}$ p/s. These promising results, when coupled with easiness of fabrication, low processing costs, and high versatility of electrode configurations, all features characterizing this manufacturing process, put into evidence lead halide perovskite, CsPbCl${}_{3}$, as a promising candidate for future radiation monitoring in proton therapy.

Basiricò et al. [85] reported on a flexible proton beam detector based on mixed 3D-2D perovskite film deposited from solution onto thin plastic foils. In particular, the 3D perovskite is MAPbBr${}_{3}$ and the 2D one is (PEA)${}_{2}$PbBr${}_{4}$. The final detector active layer is 1 μm. The device is also highly flexible with a bending radius of 3.1 mm. In Figure 12b a photo of the device is shown. The detector has been characterized under a 5 MeV proton beam with fluxes in the range [$4.5\times 10^{5}$–$1.4\times 10^{9}$] H${}^{+}$ cm${}^{-2}$ s${}^{-1}$, with repetitive irradiation cycles for over 40 min. The experimental sensitivity was found to be up to 290 nC${\mathrm{Gy}}^{-1}$${\mathrm{mm}}^{-3}$ and the detection limit as low as 72 μGy$s^{-1}$.

## 5. Conclusions

Perovskites have excellent electronic and optical properties. Together with their easily tunable composition and low-cost large-scale manufacturing processes, they have hastened in recent years the development of novel devices for ionizing radiation detection. Results reported in this review show the strong potential of this class of semiconductor material in matching several stringent requirements connected to various particle/radiation fields and to their application environments. Suitable mean ionization energy exploited for high sensitivity per unit volume/surface, bandgaps ensuring low leakage currents for low detection limit, high μτ products facilitating charge carrier transport, ability to be grown on different kind of substrates, as well as room temperature and low-voltage bias operation, easy integration on flexible devices, seem indeed to be already well-demonstrated properties. To strengthen this roadmap, forthcoming studies should be directed toward a thorough understanding of defect-assisted processes, which are influencing the performance of these devices through many parameters, such as gain, sensitivity, time response, and stability. Moreover, detailed radiation-induced defect studies should be systematically performed in order to understand radiation damage and to ensure stability under accumulated doses and ageing. Finally, as a possible extension of perovskite film-based radiation detectors beyond the application fields discussed in this review, it is worthwhile mentioning their possible evolution toward space. This may be indeed an attractive novel field of investigation, where their high versatility and detection sensitivity could offer intriguing solutions to the extreme environment conditions encountered in space. In fact, perovskite solar cells are already actively being explored as a next-generation light-weight space photovoltaic technology and their application as on-board detectors in space experiments could find their way to joining parallel efforts to this main path.

## Figures and Tables

**Figure 1 sensors-23-04930-f001:**
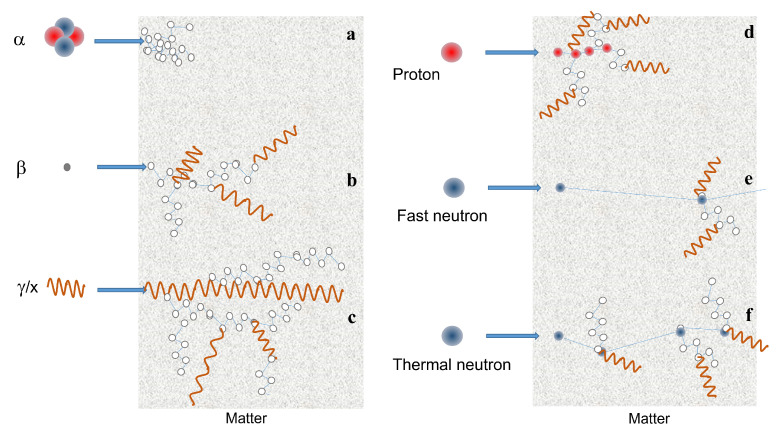
Scheme of the interaction with matter of ionizing particles: (**a**) α particle. (**b**) β particle. (**c**) x and γ-rays. (**d**) proton. (**e**,**f**) fast and thermal neutron, respectively. Neutron matter interaction probability is higher at low energies (thermal or cold neutrons), whereas high energy neutrons (fast) have only a few interactions. Thermal neutrons are much more scattered by atomic nuclei than fast neutrons, so that for the same material, the mean free path is short for thermal neutrons, whereas fast ones are able to penetrate for longer distances.

**Figure 2 sensors-23-04930-f002:**
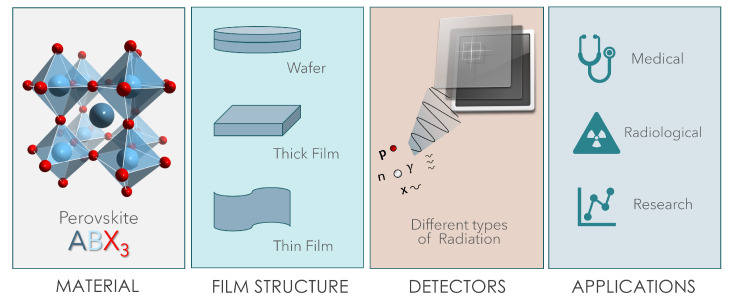
Graphic illustration depicting the topics covered in this review: starting from the left, the first panel shows the crystalline structure of a perovskite of the ABX${}_{3}$ type, the second panel is dedicated to the morphologies addressed in the review, the third schematizes the detection of different types of ionizing radiation and the fourth reports the applications of perovskite-based detectors.

**Figure 3 sensors-23-04930-f003:**
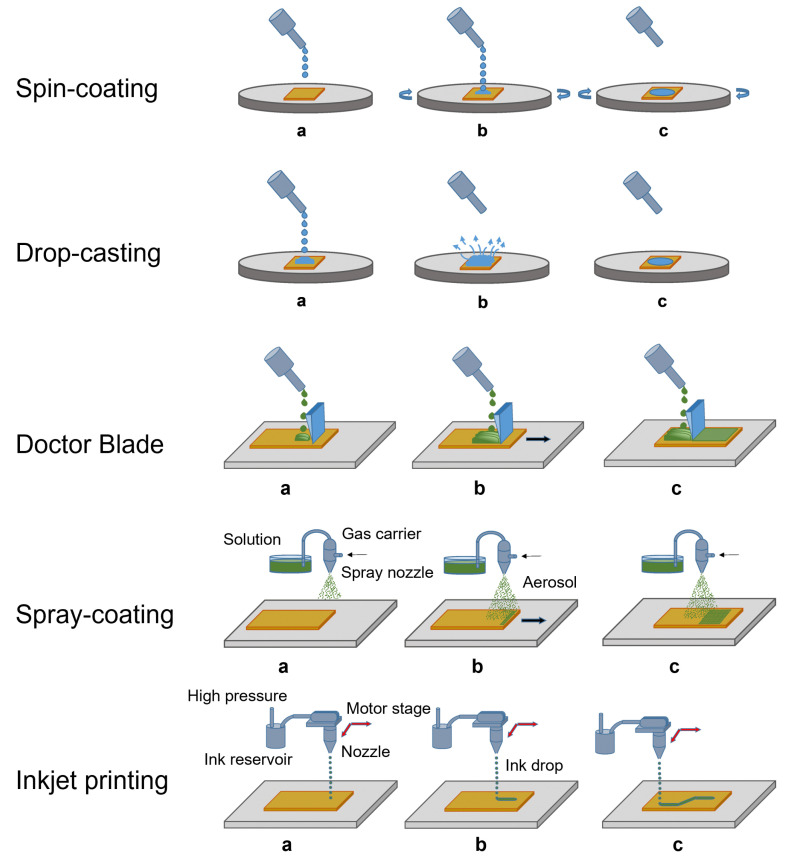
Steps of the most common solution-based processes used for the deposition of thick and thin films. (**a**) The solution is sent to the substrate. (**b**) The solution deposits on the substrate. (**c**) The film is growing.

**Figure 4 sensors-23-04930-f004:**
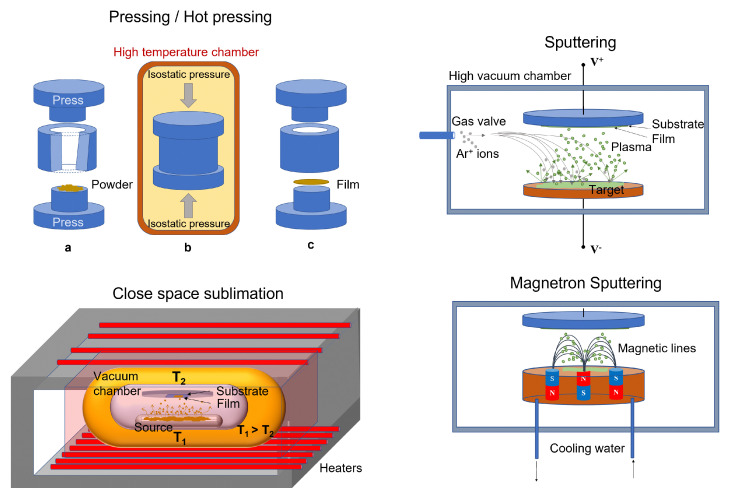
Some solution-free methods commonly used for the growth of thick and thin perovskite films as active medium for ionizing radiation detectors. Top-left: (**a**) the powder in the press before pressing; (**b**) isostatic pressing of the powder. The process can take place at high temperature; (**c**) the film is formed.

**Figure 5 sensors-23-04930-f005:**
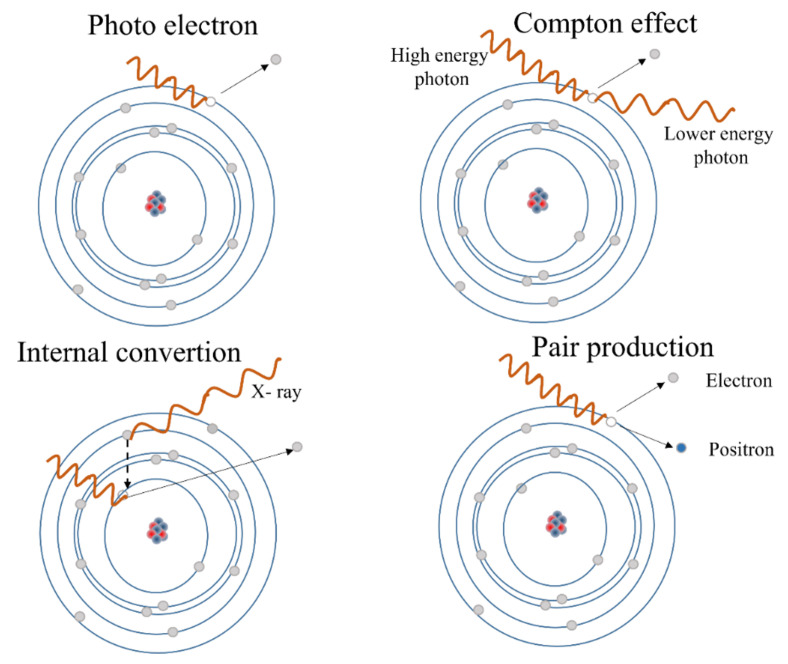
The different phenomena describing the interaction of X-rays with matter are shown: the Compton scattering, photoelectric absorption, pair production, and the internal conversion.

**Figure 6 sensors-23-04930-f006:**
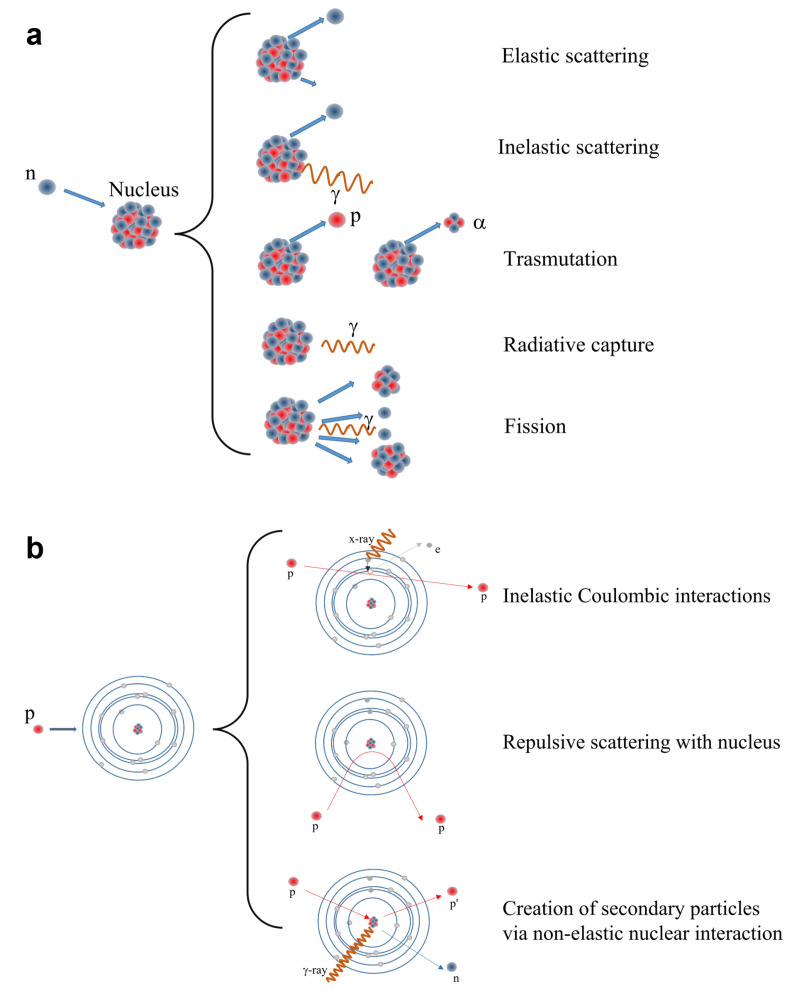
Phenomena of matter-radiation interactions for neutrons (**a**) and protons (**b**) are shown.

**Figure 7 sensors-23-04930-f007:**
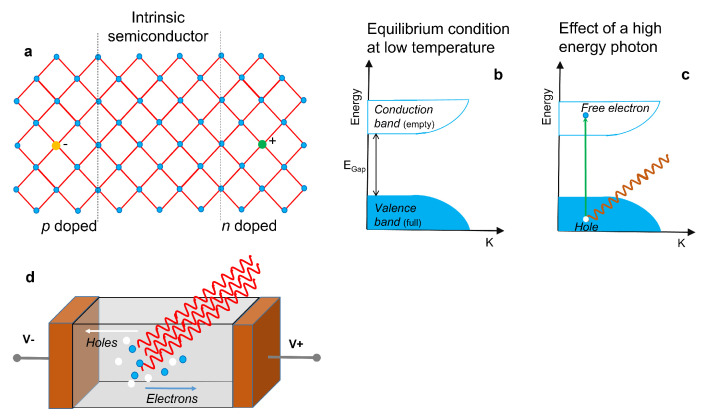
(**a**) Sketch of an intrinsic semiconductor, a p-doped and a n-doped semiconductor. (**b**) Condition of equilibrium of the semiconductor bands. (**c**) Effect of a high-energy photon incident on the semiconductor with the creation of an electron–hole pair. (**d**) Operating mechanisms of a solid-state X-ray detector, where electrons and holes are driven by the electric field and collected by the contacts.

**Figure 8 sensors-23-04930-f008:**
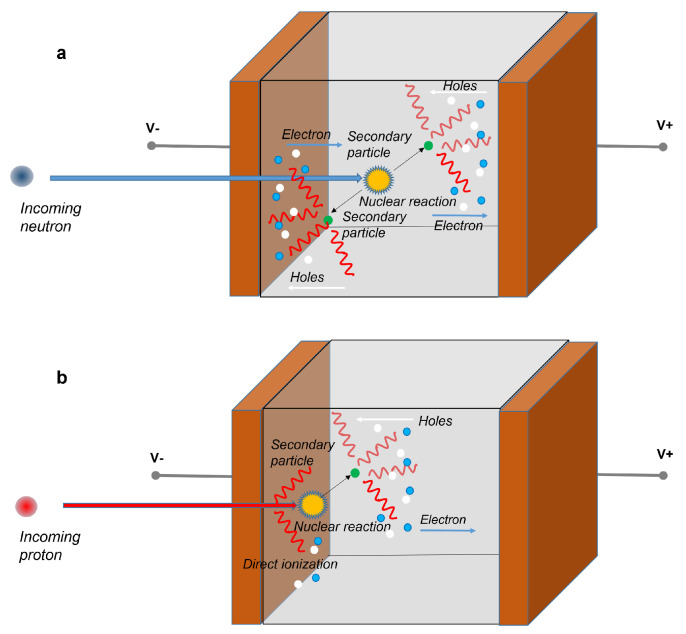
Operating mechanisms of a solid-state detector for the detection of neutrons (**a**) and protons (**b**). Secondary electrons and holes are generated by the interaction of the primary particle (neutron or proton) with the active medium of the detector. Protons can also directly ionize matter along their path. Anyhow, the so generated electrons and holes are driven by the electric field and collected by the contacts.

**Figure 9 sensors-23-04930-f009:**
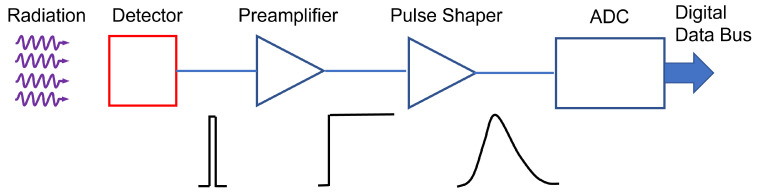
Sketch describing the steps of a basic radiation detection chain.

**Figure 10 sensors-23-04930-f010:**
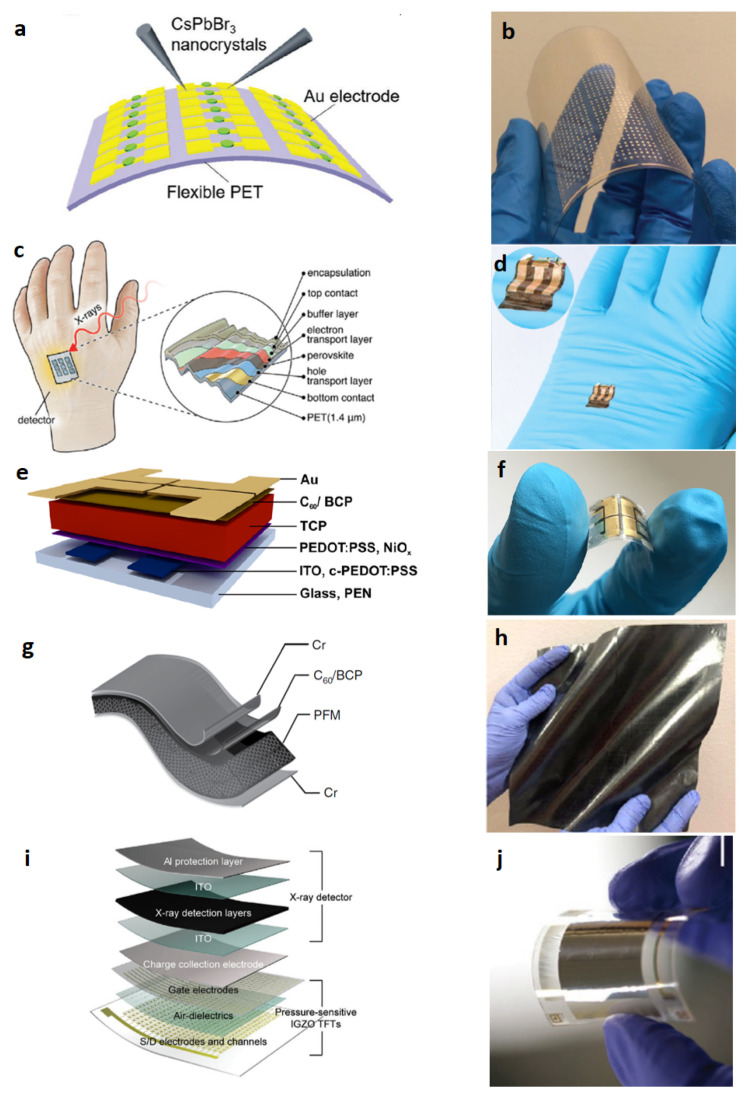
Device architectures (left) and photos (right) of flexible X-rays detectors. (**a**) Schematic of the flexible CsPbBr${}_{3}$ QDs perovskite-based detector arrays on PET substrate. (**b**) Photography of flexible CsPbBr${}_{3}$ QDs device under bending. Reproduced with permission from ref. [12]; (**c**) Schematic of the thin film perovskite ultraflexible detector fabricated on 1.4
μm PET foil. (**d**) Photograph of a detector on a laboratory glove (inset magnified twice). Reproduced with permission from ref. [66]; (**e**) Device architecture and layer stack. Detectors are realized on flexible polyethylene naphthalate (PEN) substrates Inkjet-printed layers of triple cation perovskite (TCP) are used as X-ray conversion layers. (**f**) Photograph of a flexible inkjet-printed TCP detector. Reproduced with permission from ref. [11]; (**g**) Schematic of a perovskite-filled membrane (PFM) device structure. (**h**) Photo of a large-area (400 cm${}^{2}$) flexible nylon membrane after loading perovskites. Reproduced with permission from ref. [74]; (**i**) Schematic layouts of the multiplexed detector that consisted of pressure-sensitive IGZO TFTs and X-ray detectors. (**j**) Photograph of a flexible, high-resolution multiplexed detector with a sensing area of 2 × 2 cm${}^{2}$. Reproduced with permission from ref. [73].

**Figure 11 sensors-23-04930-f011:**
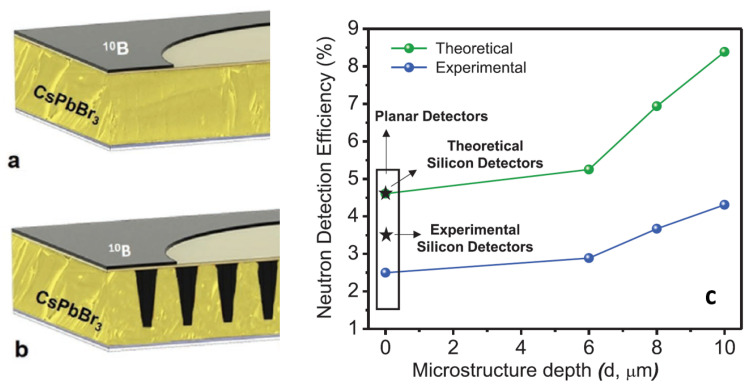
Schematic image of the planar CsPbBr${}_{3}$/Ga${}_{2}$O${}_{3}$ neutron detector (**a**) and of the microstructured CsPbBr${}_{3}$/Ga${}_{2}$O${}_{3}$ neutron detector back-filled with ${}^{10}$B (**b**). (**c**) Comparison of neutron detection efficiencies between theoretical simulations and experimental results. Reproduced with permission from ref. [83].

**Figure 12 sensors-23-04930-f012:**
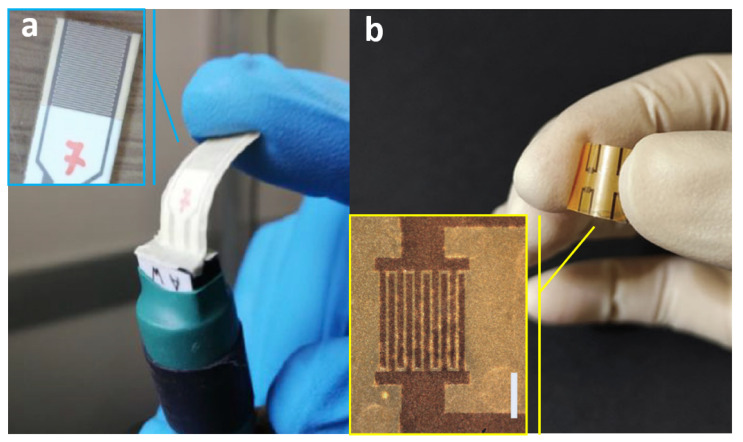
(**a**) Photos of the CsPbCl${}_{3}$ perovskite-based flexible device for radiation monitoring in proton therapy. Reproduced with permission from ref. [9]. (**b**) Photos of the mixed 3D-2D perovskite flexible proton beam detector. Reproduced with permission from ref. [85].

**Table 1 sensors-23-04930-t001:** Table showing the performance of X-rays detectors based on organic perovskites. Film growth method, thickness and detector performance details are reported.

Material	Device Structure	Growth Method	Thickness (μm)	*V* or *E*	μτ (cm${}^{2}$V${}^{-1}$)	Sensitivity (μC Gy${}^{-1}$ (cm${}^{-2}$)	Gain	X-ray Source	Year [Ref.]
MAPbI_3_	photoconductor	hydraulic press	1000	0.2 V ${\mu m}^{-1}$	$2\times 10^{-4}$	$2.527\times 10^{3}$		70 kV	2017 [39]
MAPbI_3_	photoconductor	doctor blade	830	200 V	$1\times 10^{-4}$	$1.1\times 10^{4}$		100 kV	2017 [38]
MAPbI_3_	photoconductor	hot pressing	800	10 V	$3.84\times 10^{-4}$	$1.22\times 10^{5}$	1800	40 kV	2020 [40]
MAPbI_3_	photoconductor	doctor blade	200	0.5 V ${\mu m}^{-1}$	$1.3\times 10^{-4}$	$1.35\times 10^{4}$		70 keV	2020 [69]
MAPbI_3_	p-i-n photodiode	hot pressing	570	5 V		86	411	40 kV	2022 [70]
MAPbI_3_	photoconductor	hot pressing	1300	6.58 V ${\mathrm{mm}}^{-1}$	$5.46\times 10^{-3}$	$4.2\times 10^{5}$	6126	40 kV	2021 [67]
MAPbI_3_	photoconductor	hydraulic press	880	0.17 V ${\mu m}^{-1}$	$4\times 10^{-4}$	$9.3\times 10^{3}$		50–120 kV	2021 [71]
MAPbI_3_ *	photoconductor	printing technique	10	<4 V		494		150 kV	2021 [65]
MAPbI_3_ + TMTA	Dual-Schottky barrier	doctor blade	400	500 V${\mathrm{mm}}^{-1}$	$5.3\times 10^{-5}$	$1.743\times 10^{4}$		50 kV	2022 [72]
GA_0.1_MA_0.9_PbI_3_ *	diode	doctor blade	25	0.1 V ${\mu m}^{-1}$		$6.74\times 10^{3}$		50 kV	2021 [73]
MAPb(I_0.9_Cl_0.1_)_3_ *	photoconductor	solution fill	1050	0.05 V ${\mu m}^{-1}$		$8.696\times 10^{3}$		60 keV	2020 [74]
MA_3_Bi_2_I_9_-PPs **	photoconductor	isostatic press	1000	2100 V${\mathrm{cm}}^{-1}$	$4.6\times 10^{-5}$	563		45 kV	2020 [75]

* flexible device. ** lead-free perovskite.

**Table 2 sensors-23-04930-t002:** Table showing the performance of X-ray detectors based on inorganic and multiple-cation halide perovskite (HP). Film growth method, thickness and detector performance details are reported.

Material	Device Structure	Growth Method	Thickness (μm)	*V* or *E*	μτ (cm${}^{2}$V${}^{-1}$)	Sensitivity (μC Gy${}^{-1}$ (cm${}^{-2}$)	Gain	X-ray Source	Year [Ref.]
Inorganic HP									
CsPbBr_3_	Schottky FTO/perov./Au	hot press	240	5 V${\mathrm{mm}}^{-1}$	$1.32\times 10^{-2}$	$5.568\times 10^{4}$		50 kV	2019 [42]
CsPbBr_3_	Schottky Au/perov./ITO	dissolution recrystall.	18	0.11 V ${\mu m}^{-1}$		$1.1\times 10^{3}$		35 kV	2019 [76]
CsPbBr_3_	planar Au/perov./Au photocond.	drop casting	20	5 V		***		6 MV	2019 [25]
Cs_2_TeI_6_ **	Au/perov./FTO photocond.	electrostatic spray coating	25	250 V${\mathrm{cm}}^{-1}$		19.2		40 kV	2019 [37]
CsPbBr_3_	Ga/perov./FTO Schottky	melt process	250	$1.2\times 10^{4}$ V ${\mathrm{cm}}^{-1}$		$1.45\times 10^{3}$		70 kV	2020 [43]
CsPbBr_3_	Au/perov./ITO photocond.	mist deposition	110	5 V		$1.184\times 10^{4}$		70 kV	2020 [77]
Cs_2_AgBiBr_6_ **	W/perov./Pt photocond.	mist deposition	92	109 V${\mathrm{mm}}^{-1}$		487		70 kV	2021 [78]
Cs_2_AgBiBr_6_ **	Au/perov./Au	isostatic press	1000	0.5 V ${\mu m}^{-1}$	$5.5\times 10^{-3}$	250		50 keV	2019 [41]
Cs_3_Bi_2_Br_3_I_6_ **	Cr/perov./Pt	hydraulic press	700	200 V	$1.17\times 10^{-5}$	0.4		70 kV	2021 [79]
Multiple-cation HP									
Cs_0.15_FA_0.85_PbI_3_/ Cs_0.15_FA_0.85_ Pb(I_0.15_Br_0.85_)_3_	heterojunction	solution fill	500	25 V	$8.47\times 10^{-3}$	$1.629\times 10^{3}$		60 keV	2021 [80]
PEA_2_MA_8_Pb_9_I_28_	photoconductor	doctor blade	300	600 V${\mathrm{mm}}^{-1}$	$2.6\times 10^{-5}$	$1.086\times 10^{4}$	60.43	30 keV	2022 [81]
Cs_0.1_(FA_0.83_MA_0.17_)_0.9_ Pb(Br_0.17_I_0.83_)_3_ *	photoconductor	inkjet printing	3.7	0.1 V	$2\times 10^{-6}$	59.9		70 kV	2020 [11]

* flexible device. ** lead-free perovskite. *** 70 nC${\mathrm{Gy}}^{-1}$${\mathrm{mm}}^{-3}$.

**Table 3 sensors-23-04930-t003:** Table showing solid-state devices for X-ray detection based on thin films of different kinds of perovskites. Film growth method, thickness and detector performance details are reported.

Material	Device Structure	Growth Method	Thickness (nm)	*V* (V)	μτ (cm${}^{2}$V${}^{-1}$)	Sensitivity (μC Gy${}^{-1}$ (cm${}^{-2}$)	Active Area	X-ray Source	Year [Ref.]
CsPbCl_3_	Cu/Perov./Cu	Magn. sputter	1000	10		**	1.2 × 1.8 mm${}^{2}$	6 MV	2022 [26]
CsPbBr_3_ *	Photocond. Au/Perov./Au	inkjet printing	20			17.7	0.06 ${\mathrm{mm}}^{2}$		2019 [12]
(BA)_2_(MA)_2_Pb_3_I_10_	p-i-n photod. ITO/PTAA /Perov./C60/Au	AVC	470	0.5		13	0.04 ${\mathrm{cm}}^{2}$	10.91 k eV	2020 [82]
Cs_0.05_FA_0.79_MA_0.16_ Pb(I_0.8_Br_0.2_)_3_	p-i-n photod.	spin coating	450	0.4	$2\times 10^{-5}$	97	2 × 3 ${\mathrm{mm}}^{2}$	35 keV	2019 [35]
MAPbI_3_	Planar photocond.	spin coating	260–600		$2\times 10^{-7}$	16–20		75 kV	2015 [24]
Cs_0.05_(FA_0.83_MA_0.17_)_0.95_ PbI_3-x_Br_x_ *	p-i-n photod. ***	spin coating	500	1		9.3	0.05 ${\mathrm{cm}}^{2}$	15.2 k eV	2020 [66]

* flexible substrate. ** 640 nC${\mathrm{Gy}}^{-1}$${\mathrm{mm}}^{-3}$ *** PEDOT/Perov./PTCDI/Cr${}_{2}$O${}_{3}$ (Large Contact).

**Table 4 sensors-23-04930-t004:** Main performances of thermal neutron detectors based on perovskite films.

Material	Device Structure	Growth Method	Detection Mechanism	Neutron Source	Detection Efficiency	Year [Ref]
CsPbBr_3_	ITO/Ga_2_O_3_/CsPbBr_3_/Au	close space sublimation	indirect	^252^Cf	2.5%	2020 [44]
CsPbBr_3_	ITO/Ga_2_O_3_/CsPbBr_3_/Au	close space sublimation	indirect	^252^Cf	4.3%	2022 [83]

## Data Availability

Data sharing not applicable.

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
