# Peer review of "Halide Perovskites Films for Ionizing Radiation Detection: An Overview of Novel Solid-State Devices"

_sensors, 2023, doi:10.3390/s23104930_

Round 1

Reviewer 1 Report

The quality of english language is good; I suggest to check some possible inconsistency between American and British english, selecting one of them for the whole paper.

Reviewer 2 Report

The paper

Halide Perovskites Films for Ionizing Radiation Detection: An Overview of Novel Solid-State Devices

By Naomi Falsini  et al.

Provides a clear and comprehensive review of the fabrication and application of Halide Perovskites Films as ionizing radiation detectors. There are some overlaps with other reviews [13-15] relevant to this field, however, they are correctly cited and figures are reproduced with permission.

The manuscript is well-written, clear, and well-structured. Being a review paper, the bibliography is extensive, to my best knowledge, exhaustive, and does not include an excessive number of self-citations (8/75).

The paper can be accepted for publication in Sensors journal, subject to minor revisions based on the following comments and recommendations:

Row 38: …for non-destructive investigation of materials and textiles. Why textiles, are they to be intended as biological tissues?

Figure 2: a legend for the first scheme could be helpful to locate the A,B,X components

Row 86: …. for applications in the field of light generation (LEDs [16–19] and lasers [20], missed closing parenthesis.

Row 169:” The uniformity of the film and its thickness depends on the deposition parameters, such as the rotation speed, but also on the concentration of the solution and its density”. Viscosity play also a key role.

Section 3.1: it could be helpful to mention if the solvent removal occurs at room temperature or if heat treatments can be adopted.

Row 359: The recoil nucleus, which in the case of hydrogen is the proton, behaves like an α particle transferring its energy to the surrounding electrons. Some more comments are required to clarify the statement “behaves like an alpha particle”.

Row 396: the paper does not deal with scintillators, so it seems to me that the paper deals with electronic devices (first category).

Row 420: the signal emerging from the detection of a single particle is not due to the collection of charge carriers generated by ionization, but from the charge induced at the sensing electrode by the motion of the charge carriers (as in S. Del Sordo et al. Sensors 2009, 9, 3491-3526). This basic concept should be underlined in the text, corroborated by some references.

Eq. (3). It is not clear why mu and tau refer only to majority carrier mobility and lifetime. Both carriers contribute to the photocurrent.

Eq. (4). To my best knowledge, the Hecht formula is valid only if the electric field is constant and if the generation depth is much smaller than the device thickness. In this case, the signal is mainly given by one carrier. Authors should clarify the conditions of application of such a model.

Reviewer 3 Report

Very interesting topic and well written manuscript. There are only minor comments relating to the paper:

-In Figure 9. the pulse shape after the preamplifier seems to be wrong, as an amlified pulse should remain pulse-like 

-It would be interesting to talk about the applicability of perovskite films in space applications, e.g. for onboard detectors, maybe in a future extension of the article.

-Please review the using of capital letter for intersentence words; it is not consistent in the text.
